# Integrating Serum Biomarkers into Prediction Models for Biochemical Recurrence Following Radical Prostatectomy

**DOI:** 10.3390/cancers13164162

**Published:** 2021-08-19

**Authors:** Shirin Moghaddam, Amirhossein Jalali, Amanda O’Neill, Lisa Murphy, Laura Gorman, Anne-Marie Reilly, Áine Heffernan, Thomas Lynch, Richard Power, Kieran J. O’Malley, Kristin A. Taskèn, Viktor Berge, Vivi-Ann Solhaug, Helmut Klocker, T. Brendan Murphy, R. William Watson

**Affiliations:** 1School of Mathematical Sciences, University College Cork, T12XF62 Cork, Ireland; 2UCD School of Medicine, Conway Institute of Biomolecular and Biomedical Research, UCD, D04V1W8 Dublin 4, Ireland; Amanda.oneill@ucd.ie (A.O.); lisa.murphy24@mail.dcu.ie (L.M.); laura.gorman@ucdconnect.ie (L.G.); annemarie.reilly44@mail.dcu.ie (A.-M.R.); aine.heffernan@ucdconnect.ie (Á.H.); william.watson@ucd.ie (R.W.W.); 3Department of Urology, Trinity College, St James Hospital, D08 W9RT Dublin 8, Ireland; tlynch@stjames.ie; 4Department of Urology, Royal College of Surgeons in Ireland, Beaumont Hospital, D09V2N0 Dublin 9, Ireland; rpower@rcsi.ie; 5Department of Urology, University College Dublin, Mater Misericordiae University Hospital, D07YH5R Dublin 7, Ireland; kiaranomalley@mater.ie; 6Institute of Clinical Medicine, University of Oslo, 0318 Oslo, Norway; k.a.tasken@medisin.uio.no (K.A.T.); viktor.berge@medisin.uio.no (V.B.); 7Department of Tumor Biology, Oslo University Hospital, 0379 Oslo, Norway; 8Department of Urology, Oslo University Hospital, 0379 Oslo, Norway; Vivi-Ann.Solhaug@nordlandssykehuset.no; 9Department of Urology, Medical University of Innsbruck, 6020 Innsbruck, Austria; helmut.klocker@i-med.ac.at; 10UCD School of Mathematics and Statistics, University College Dublin, D04V1W8 Dublin 4, Ireland; brendan.murphy@ucd.ie

**Keywords:** biochemical recurrence, calibration, Cox model, discrimination, model evaluation, prediction models, prostate cancer, cytokine

## Abstract

**Simple Summary:**

Treatment decisions represent a significant dilemma for patients diagnosed with prostate cancer. The prediction of early treatment failure would inform appropriate decision making and allow the clinician and patient to consider appropriate primary treatments and adjuvant therapies. We have developed and validated a serum biomarker-based model for predicting risk of biochemical reoccurrence in prostate cancer after radical prostatectomy. This study shows that the pre-operative biomarker PEDF can improve the accuracy of the clinical factors to predict risk of biochemical reoccurrence. PEDF has anti-inflammatory effects impacting on cytokine production. This non-invasive tool can be employed prior to treatment and demonstrates significant benefit over current clinical practice, impacting on patients’ outcomes and quality of life.

**Abstract:**

This study undertook to predict biochemical recurrence (BCR) in prostate cancer patients after radical prostatectomy using serum biomarkers and clinical features. Three radical prostatectomy cohorts were used to build and validate a model of clinical variables and serum biomarkers to predict BCR. The Cox proportional hazard model with stepwise selection technique was used to develop the model. Model evaluation was quantified by the AUC, calibration, and decision curve analysis. Cross-validation techniques were used to prevent overfitting in the Irish training cohort, and the Austrian and Norwegian independent cohorts were used as validation cohorts. The integration of serum biomarkers with the clinical variables (AUC = 0.695) improved significantly the predictive ability of BCR compared to the clinical variables (AUC = 0.604) or biomarkers alone (AUC = 0.573). This model was well calibrated and demonstrated a significant improvement in the predictive ability in the Austrian and Norwegian validation cohorts (AUC of 0.724 and 0.606), compared to the clinical model (AUC of 0.665 and 0.511). This study shows that the pre-operative biomarker PEDF can improve the accuracy of the clinical factors to predict BCR. This model can be employed prior to treatment and could improve clinical decision making, impacting on patients’ outcomes and quality of life.

## 1. Introduction

Radical prostatectomy (RP) represents a curative intent for localised prostate cancer; however, 20–30% of patients will fail, developing biochemical recurrence (BCR) [1]. The pre-treatment prediction of early treatment failure would inform appropriate decision making and allow the clinician and patient to consider alterative primary treatments and adjuvant therapies. Current clinical variables to inform outcome are pre-treatment PSA, biopsy Gleason Score (bxGS) and clinical stage, but additional biomarkers are needed to improve on their prediction. Commercially available tests for the prediction of BCR include Oncotype DX, Prolaris and Decipher, but these are dependent on tissue biopsies, introducing a sampling error [2,3].

Serum or urine biomarkers would overcome this and represent a less invasive approach to predicting BCR. To date, limited studies have identified serum biomarkers for predicting BCR. Svatek et al. has identified preoperative biomarkers that improve the accuracy of standard models associated with BCR after RP [4]. This supports the concept that panels of blood-based proteins could improve on the prediction of BCR.

Prediction models have been widely used to aid clinical decision making. They mainly are mathematical equations or algorithms that assign a level of risk using patient variables. There are a range of time-to-event models that predict the duration of time expected before an event (e.g., recurrence of a disease) occurs rather than predicting the risk of the event at a specific time point (e.g., in logistic regression). The Cox proportional hazards model [5] is the most commonly used regression model for modelling time-to-event data.

In our current study we undertook to investigate the ability of a panel of biomarkers to predict BCR and build a model integrating the best biomarkers with the current clinical variables that significantly improved on the current clinical decision-making tools to predict 3- and 5-year BCR. These biomarkers were selected from previous discovery studies carried out in our laboratory [6] and have evidence for a role in cancer development. Insulin-Like Growth Factor Binding Protein-3 (IGFBP-3) is a p53 tumour suppressor-regulated protein and binds to IGF1, facilitating its transport in the circulation. Disruption of IGFBP-3 at the transcriptional and post-translational levels has been implicated in the pathophysiology of many cancers, including that of the prostate [7]. Apolipoprotein A-II (APOAII) has been shown to be associated with pancreatic cancer [8]. APOAII is overexpressed in prostate cancers and may be involved in cell proliferation and apoptosis [9]. CD14 is an integral part of the innate immune system. CD14-positive tumours were shown to be more vascularised in certain cancers, such as bladder cancer [10]. It has also been shown to be released into the circulation and promote tumour-related inflammation. Vitamin D-Binding Protein (VDBP/Gc-Globulin) binds Vitamin D and has been shown to increase the risk of prostate cancer [11]. Zinc α2-glycoprotein (ZAG) has been identified by proteomics in prostate tissue and urine, showing different concentrations in patients with prostate cancer and benign prostate hyperplasia [12]. Pigment Epithelium-Derived Factor (PEDF) has been identified as a mediator of inflammation and regulates macrophage activation [13]. It has been shown to stimulate macrophages to release tumour necrosis factor and interleukin-1 [13] as well as inducing the migration of monocytes and macrophages [14], contributing to the maintenance of chronic inflammation. Decreased Tumour-Associated Macrophages is associated with prostate cancer progression [15]. Loss of PEDF could lead to reduced immunological effects and lead to the progression of the tumour, leading to BCR.

The developed model was then internally and externally validated to evaluate its predictive performance and avoid overfitting [16].

## 2. Patients and Methods

### 2.1. Patient Population and Sample Collection

Three radical prostatectomy (RP) cohorts were used (Table 1). The Norwegian cohort was a selected group of patients consisting of an equal number of organ-confined and non-organ-confined samples and Gleason 6, 7 and 8. The majority of men in all three cohorts were Caucasian; no patients received neoadjuvant hormonal therapy or had a prior history of pelvic radiotherapy, which were the exclusion criteria. The inclusion criteria for this study included the availability of a preoperative serum sample, PSA, transrectal ultrasound guided needle biopsy pathology reports and the corresponding radical prostatectomy pathology report. BCR was defined as two consecutive PSA values >0.4 ng/mL, which correlate best with clinical progression, metastasis and need for adjuvant therapy [17].

Blood samples were collected within three months prior to their RP and serum was isolated and stored at −80 °C until analysis. All samples were transported to the Conway Institute for analysis using established standard operating procedures. See ethical approval and consent details below.

### 2.2. Biomarker Measurements

A multiplex antibody-based electrochemiluminescence assay was developed (Meso Scale Discovery (MSD), Gaitherberg, MD, USA). Custom-made plates were generated. Five serum-based biomarkers were assessed on a 5-plex, 96-well 7-Spot MSD microplate plate: APOAII (Apolipoprotein AII, Abcam; ab20903, Cambridge, UK), CD14 (Cluster Differentiation 14; RnD System DY383, Abingdon, UK), Gc-globulin (Vitamin D-Binding Protein; ThermoFisher HYB 249-01B-005, Dublin, Ireland), IGFBP-3 (Insulin-Like Growth Factor Binding Protein 3 DuoSet ELISA; RnD System DY675, Abingdon, UK) and ZAG (Zinc-alpha-2-glycoprotein; Biotechne BAF4764, Dublin, Ireland). The second panel consisted of two serum-based biomarkers: PEDF (Pigment Epithelium-Derived Factor; MSD W0060165) was assessed on the Duplex 96-well 4-Spot MSD microplate plate and PSA (prostate-specific antigen; MSD W0016950, Rockville, MA, USA), following on from previous studies in our laboratory [18]. The average detection ranges for each biomarker were APOAII = 480.2–1250,000 pg/mL, CD14 = 61.93–58,500 pg/mL, Gc-Globulin = 87.2–1250,000 pg/mL, IGFBP-3 = 250–31,000 pg/mL and ZAG = 1613–250,000 pg/mL. The average detection ranges for the 2 biomarkers were as follows: PEDF = 6.21–500,000 pg/mL and PSA = 230.25–100,000 pg/mL. Internal pooled reference serum samples were also run. Samples were run as previously described [18] and plates were read on an MDS plate reader (MESO Quickplex S120 microplate reader using the MSD Workbench Software V2 (Rockville, MA, USA).

### 2.3. Statistical Method

Patient information used to build the model included the clinical risk factors (PSA, Digital rectal examination (DRE) and biopsy Gleason Score (bxGS)) and serum biomarkers identified previously by our group (CD14, IGFBP-3, APOAII, Gc-globulin, ZAG and PEDF) [6]. The stepwise selection technique using R software [19] was applied to identify the serum biomarkers that could predict time-to-BCR. A Cox regression model was used to predict time-to-BCR using linear and nonlinear effects of the serum biomarkers.

The model discrimination is demonstrated using the Receiver Operator Characteristic (ROC) curve and decision curve analysis [20] at 3 and 5 years. The discriminant ability of the models was also numerically determined using the area under the ROC curve (AUC). Comparison of AUC values took place via a method described by DeLong et al. [21].

The model calibration was tested using the Greenwood–Nam–D’Agostino goodness-of-fit test [22]. Calibration plots were used to investigate how close the predicted probabilities are to the actual probabilities using Frank Harrell’s method [23]. This method used resampling techniques to get bias-corrected estimates of the predicted versus observed values for Cox models.

Internal validation using the Irish cohort was performed using 10-fold cross-validation to confirm that no patient was used to both develop and test the model. External validation was performed to evaluate the model performance in the independent Austrian and Norwegian cohorts. The validation technique for the Cox models proposed by Royston [24] was used, which adjust the baseline survival function to the new cohorts. This adjustment is crucial to assess the calibration of survival probabilities in the validation datasets since the event probabilities are estimated relative to an unspecified baseline function [24].

Due to the short follow-up time for the large number of validation cases, the external validation was only investigated at 3-year BCR. A second approach investigated a combined dataset used to provide more validation sample cases. We combined the Irish, Austrian and Norwegian cohorts and randomly selected 70% (346 samples) to generate a training set for model development, and the remaining 30% (145 samples) to test the model using the caret R package [25]. This approach provides sufficient data for model development and validation at 3- and 5-year BCR. The results of the combined cohort are presented in the Appendix A and described in the results section, as additional prospective studies would be required to further validate these findings.

## 3. Results

The model for predicting BCR after RP was developed using the Irish cohort (271 samples) and validated in the independent Austrian (128 samples) and Norwegian (178 samples) cohorts. Figure 1 represents the cumulative probability of time-to-BCR by cohort.

The clinical characteristics grouped by cohort is shown in Table 1. The rate of 3-year BCR across the three cohorts are 15% for Irish, 29.3% for Austrian and 28.7% for Norwegian, and 18.5% for Irish, 37.8% for Austrian and 75.5% for Norwegian across the first 5 years of follow-up. It should be noted that the Norwegian cohort have a higher rate of 5-year BCR because they are a selected cohort, as previously published by Berge et al., for which the rate of 5-year BCR is 27% [26]. The *p*-values for each clinical variable (Table 1) indicate significant differences in the clinical variables (except stage) between the three cohorts.

The effect of the panel of biomarkers (APOAII, CD14, Gc-globulin, IGFBP-3 and ZAG) combined with the clinical risk factors was investigated using the Cox model for the Irish cohort. The variable selection technique identified the best combination of biomarkers to integrate with either the classic three-tier ‘NCCN’ risk groups (low, intermediate and high) (‘NCCNbio’) or the ‘Clinical’ variables (PSA, DRE and bxGS) (‘Clinicalbio’), respectively. The effects of the risk factors on time-to-BCR for both models are given in Table 2.

The ‘NCCNbio’ model indicates that the intermediate-risk NCCN group and high-risk NCCN group are expected to have 1.8- and 3.1-times higher risk of developing BCR over time compared to the low-risk group, respectively. A 100,000 (pg/mL) increase in CD14 was also shown to increase the risk of BCR by 2%, while a 100,000 (pg/mL) increase in PEDF was shown to decrease the risk of BCR by 17%.

The ‘Clinicalbio’ model indicates that a one-unit (ng/mL) increase in the logarithm of PSA was shown to increase the risk of BCR by 163%, which is approximately a 2-fold increase. The use of log transformation for PSA was shown to be essential in our previous studies [27], where a change in smaller values of PSA is more critical. Patients with abnormal DRE were shown to have a 23% higher risk of developing BCR over time. Moreover, patients with a bxGS of 7 and 8 were shown to have a 1.5- and 3-times higher risk of BCR compared to patients with a Gleason score of 6, respectively. PEDF is also identified as an essential marker in the ‘NCCNbio’ model where a 100,000 (pg/mL) increase in PEDF was shown to decrease the risk of developing BCR by almost 20%.

Models with the NCCN risk groups (‘NCCN’) and the clinical variables, namely, PSA, DRE and bxGS (‘Clinical’), were used as references for the ‘NCCNbio’ and ‘Clinicalbio’ models, respectively. A model including the effect of PEDF alone (‘Biomarker’) was also included as an additional reference. Using cross-validation, the internal discriminate ability of both models was compared numerically (Table 3A) and graphically (Appendix A) to their references at 3- and 5-year BCR.

The AUC for the 3- and 5-year BCR (Table 3A) for the ‘NCCNbio’ model (AUC of 0.7058 and 0.6968) demonstrates a significant improvement over the ‘NCCN’ (AUC of 0.5335 and 0.5424) model alone. The ‘Clinicalbio’ model (AUC of 0.7076 and 0.7531) also demonstrates a significant improvement over ‘Clinical’ (AUC of 0.6377 and 0.6777) at 3 and 5-year BCR. The ROC (Appendix A) and decision curves (Appendix A) show that the integration of the biomarkers with the clinical factors not only increased sensitivity and specificity in almost all thresholds but also provided additional clinical benefit.

The Greenwood–Nam–D’Agostino test for ‘NCCNbio’ (*p*-value of 0.369 at 3 years and 0.610 at 5 years) and ‘Clinicalbio’ (*p*-value of 0.897 at 3 years and 0.691 at 5 years) indicate that both models are well calibrated (Appendix A).

Using the independent Austrian and Norwegian cohorts, the external discrimination ability of the models compared to their references at 3 years is shown in Table 3B and Figure 2. The AUC of the ‘NCCNbio’ model (0.7065 and 0.6224) has no significant improvement over ‘NCCN’ (AUC of 0.6958 and 0.5838) for the Austrian and Norwegian cohorts, respectively (Table 3B). However, the integration of the biomarker with clinical factors in the ‘Clinicalbio’ model (AUC of 0.7659 and 0.5877) demonstrated a significant improvement over ‘Clinical’ (AUC of 0.6971 and 0.5174). The ROC and decision curves for the Austrian cohort on the top of Figure 2A,B demonstrates that the integration of PEDF with the clinical risk factors (‘Clinicalbio’) increased sensitivity and specificity in almost all thresholds and also provided additional clinical benefits (compared to the ‘Clinical’ model).

The Greenwood–Nam–D’Agostino test for ‘NCCNbio’ (*p*-value of 0.286 of Austrian and 0.008 of Norwegian) and ‘Clinicalbio’ (*p*-value of 0.121 of Austrian and <0.001 of Norwegian) shows that both models are well calibrated to predict BCR for the Austrian cohort but not for the Norwegian cohort. The calibration plots (Figure 2C) show that both models generally give an overestimate for the chance of time-to-BCR at 3 years for the Norwegian cohort.

We also undertook to combine the three cohorts (Irish, Austrian and Norwegian) to give a larger sample size and build a separate model for predicting BCR. The model was developed using 70% of the combined cohort (405 samples) and validated using the remaining 30% of the combined cohort (172 samples). The clinical characteristics are shown in Appendix A. The *p*-values of the combined cohort in Appendix A show no significant differences in clinical variables between test and train set (except for ‘Gleason Grade’).

The model summaries for the combination of the selected biomarkers with either the classic three-tier ‘NCCN’ risk groups (‘NCCNbio-s’) or the ‘Clinical’ variables (‘Clinicalbio-s’) are given in Appendix A.

The discrimination ability of the combined cohort is shown in Appendix A. This demonstrates a significant improvement in the AUC values for ‘NCCNbio-s’ (AUC of 0.7329 and 0.7601) in comparison to ‘NCCN-s’ and ‘Clinicalbio-s’ (AUC of 0.7713 and 0.8332) in comparison to ‘Clinical-s’ for both 3- and 5-year BCR. The ROC and decision curves for the combined cohort in Appendix A show that the integration of the biomarkers with clinical factors has increasing sensitivity and specificity and provides additional clinical benefit. The calibration plots also indicate that ‘NCCNbio-s’ (*p*-value of 0.168 and 0.457) and ‘Clinicalbio-s’ (*p*-value of 0.856 and 0.716) are well calibrated at 3 and 5 years.

The external validation of the combined cohort also demonstrates improved discrimination ability. The ROC and decision curves for the test set in Appendix A show that the integration of PEDF (which was the biomarker identified using the Irish cohort to build the model and then validated in the Austrian and Norwegian cohorts, as above) into the clinical risk factors (‘Clinicalbio-s’) increased the sensitivity and specificity, and also provided additional clinical benefit. The Greenwood–Nam–D’Agostino test results for ‘NCCNbio-s’ (*p*-value of 0.335 and 0.107) and ‘Clinicalbio-s’ (*p*-value of 0.258 and 0.055) indicate that both models are calibrated in the test set at 3 and 5 years (Appendix A).

## 4. Discussion

We have developed statistical models that combine biomarkers into the clinical variables to predict BCR. The clinical information was considered on their own (PSA, DRE and bxGS) and also combined into the 3-tier NCCN [28], so they can be used as a tool for treatment decision making.

The use of survival methods and the Cox model [5] were appropriate modelling approaches. However, the initial analysis using logistic regression for 3-year and 5-year BCR [29] identified PEDF as the important marker, which was in line with the finding of the Cox model.

The methodology for validation of the Cox models was not straightforward as would be for logistic models. This is mainly because the un-estimated baseline survival in the Cox model is a vital component for the external validation. Royston [24] proposed an approach to tackle this issue, which was used for our external validation.

Traditional biomarker studies build a model in one cohort and validate in an independent cohort [30], which was the approach we undertook. However, due to the lack of 5-year follow-up in the individual cohort approach, we also undertook a combination of the cohorts to generate a combined-cohort approach. Pooling data across the cohorts helps to achieve a larger sample cohort, which allowed model validation at 5-year BCR [31,32]; however, additional prospective studies of these biomarkers will be required.

Our study demonstrated that the combination of PEDF with PSA, DRE and bxGS improved the prediction of post-operative BCR in both the individual cohort and combined-cohort studies. Previous studies by our group and others have demonstrated that the integration of serum biomarkers improves prediction models [33,34,35]. PEDF has multiple biological actions and is expressed by prostate epithelial and stromal cells [36]. Downregulation of PEDF expression in prostate cancer has been linked to poor prognosis [37] and the tumours metastatic phenotype [38]. It has been identified as a major antimetastatic factor [39], which supports its role as a predictor of disease progression.

Robust methods, including the ROC curve, calibration and decision curve analysis, were used to access model performance. The discrimination ability of the models is presented at different thresholds; however, an optimal threshold needs to be chosen to make the best clinical decision in practice. The selection of this threshold could be challenging as it depends on a trade-off between a more sensitive or a more specific test [40]. Further validation studies will be required to identify the best clinically accepted thresholds.

## 5. Conclusions

This study shows that the pre-operative biomarker PEDF can improve the accuracy of the clinical factors to predict BCR. The clinical and serum biomarker model can be employed prior to treatment and could improve clinical decision making for the physician and patient to choose the appropriate treatment, and this could impact on patients’ outcomes and quality of life.

## Figures and Tables

**Figure 1 cancers-13-04162-f001:**
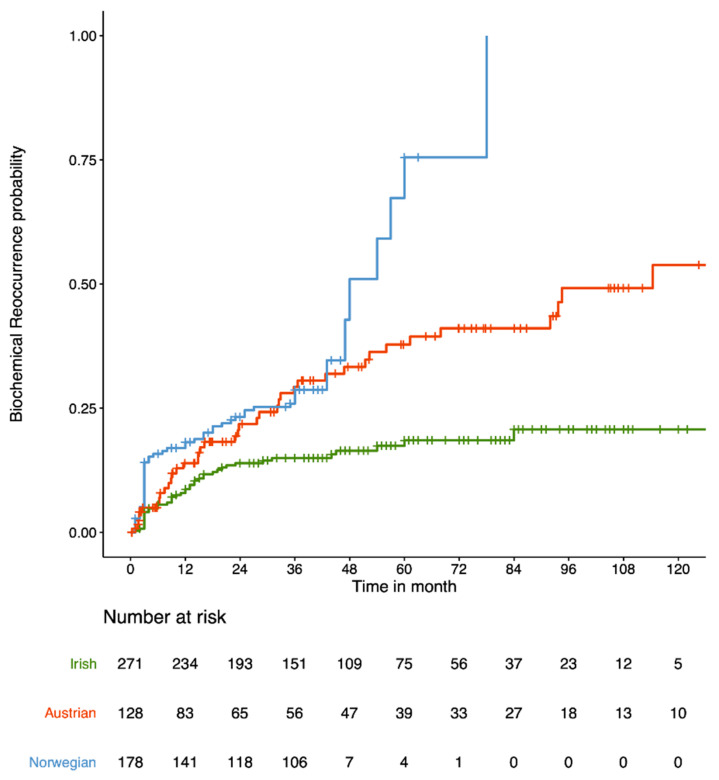
Cumulative probability of time-to-BCR (in months) for the 3 cohorts under investigation: Irish (green), Austrian (red) and Norwegian (blue).

**Figure 2 cancers-13-04162-f002:**
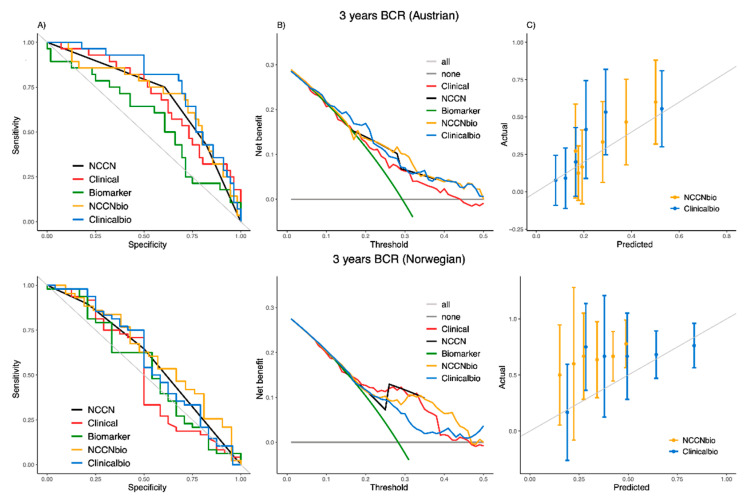
ROC curve (**A**), decision curve (**B**) and calibration plot (**C**) of the ‘NCCNbio’ and ‘Clinicalbio’ models externally compared with ‘NCCN’, ‘Clinical’ and ‘Biomarker’ models in independent Austrian (on the top) and Norwegian (on the bottom) cohorts at 3 years.

**Table 1 cancers-13-04162-t001:** Clinical characteristics of the patients by cohorts, and the univariate *p*-value for each clinical variable ^a^.

Features	Irish	Austrian	Norwegian	*p*-Value ^b^
Sample size (*n* = 577)	271	128	178	
**Pre-op**				
PSA				<0.001
Mean (SD)	8.36 (4.74)	5.77 (4.73)	11.3 (7.25)	
DRE				<0.001
Normal	198 (73%)	107 (84%)	165 (93%)	
Abnormal	73 (27%)	21 (16%)	13 (7%)	
Biopsy Gleason Score				<0.001
6	132 (49%)	73 (55%)	56 (31%)	
7	98 (36%)	47 (35%)	76 (43%)	
8 and above	42 (15%)	13 (10%)	46 (26%)	
**Post-op**				
Gleason Score				<0.001
6	92 (34%)	37 (29%)	60 (34%)	
7	135 (51%)	79 (62%)	59 (33%)	
8 or above	40 (15%)	12 (9%)	59 (33%)	
Stage				0.13
Organ-confined	155 (57%)	78 (61%)	89 (50%)	
Non-organ-confined	116 (43%)	50 (39%)	89 (50%)	
Time to biochemical recurrence				<0.001
<3 years	15.0%	29.3%	28.7%	
<5 years	18.5%	37.8%	75.5%	

^a^ Statistics presented: mean (SD) or *n* (%). ^b^ Statistical tests performed: Kruskal–Wallis test; chi-square test of independence; log-rank test.

**Table 2 cancers-13-04162-t002:** Summary of the ‘NCCNbio’ and ‘Clinicalbio’ models developed for the Irish cohort using the hazard ratio, 95% confidence interval (CI) for the hazard ratio and *p*-value for the risk factors in the model.

Features	NCCNbio			Clinicalbio		
	Hazard Ratio	95% CI	*p*-Value	Hazard Ratio	95% CI	*p*-Value
PSA ^a^	-	-	-	2.628	(1.45, 4.75)	0.001
DRE						
(Abnormal vs. normal)	-	-	-	1.227	(0.62, 2.43)	0.556
Biopsy Gleason Score						
(7 vs. 6)	-	-	-	1.516	(0.73, 3.15)	0.265
(8 or above vs 6)	-	-	-	2.99	(1.35, 6.65)	0.007
NCCN						
(Intermediate vs. low)	1.808	(0.72, 4.54)	0.207	-	-	-
(High vs. low)	3.135	(1.33, 7.39)	0.009	-	-	-
CD14 (100,000 pg/mL)	1.02	(0.99, 1.05)	0.1	-	-	-
PEDF (100,000 pg/mL)	0.831	(0.70, 0.98)	0.03	0.801	(0.68, 0.95)	0.009

^a^ The non-linear effect of the predictor using a log transformation.

**Table 3 cancers-13-04162-t003:** The AUC for the ‘NCCNbio’ and ‘Clinicalbio’ compared to ‘NCCN’, ’Clinical’ and ‘Biomarker’ for the internal validation using the Irish cohort (Panel A) and external validation using the Austrian and Norwegian cohorts (Panel B).

Models	(A) Internal Validation		(B) External Validation	
	AUC at 3-Year (Irish Cohort)	AUC at 5-Year (Irish Cohort)	AUC at 3-Year (Austrian Cohort)	AUC at 3-Year (Norwegian Cohort)
NCCN	0.5335	0.5424	0.6958	0.5838
Clinical	0.6377	0.6777	0.6971	0.5174
Biomarker	0.5928	0.6236	0.5702	0.5330
NCCNbio	0.7058; *p*-value (vs. NCCN) < 0.001 ^a^	0.6968; *p*-value (vs. NCCN) = 0.002 ^a^	0.7065; *p*-value (vs. NCCN) = 0.901 ^a^	0.6224; *p*-value (vs. NCCN) = 0.701 ^a^
Clinicalbio	0.7076; *p*-value (vs. Clinical) = 0.024 ^a^	0.7531; *p*-value (vs. Clinical) = 0.032 ^a^	0.7659; *p*-value (vs. Clinical) = 0.034 ^a^	0.5877; *p*-value (vs. Clinical) = 0.042 ^a^

^a^ DeLong test *p*-value.

## Data Availability

The data are available to other researchers on written request to the corresponding author.

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
