# Peer review of "Integrating Serum Biomarkers into Prediction Models for Biochemical Recurrence Following Radical Prostatectomy"

_cancers, 2021, doi:10.3390/cancers13164162_

Round 1

Reviewer 1 Report

The goal of the study performed was to predict Biochemical Recurrence (BCR) in prostate cancer patients after radical prostatectomy using serum biomarkers. The authors investigated the ability of a panel of biomarkers and have developed and validated a serum biomarker-based model for predicting risk of biochemical reoccurrence in prostate cancer. They found that the combination of Pigment epithelium-derived factor (PEDF) with other markers, such as PSA, DRE and bxGS, improved on the prediction of post-operative BCR in both the individual cohort and combined cohort studies.

This is an interesting and promising finding. The manuscript can be accepted for publication.

Minor comment.

  1. Probably, deciphering PEDF in the abstract, as it has been done for Biochemical Recurrence (BCR), makes sense.
  2. In Fig. 2, the font of the labels is too small.

Author Response

  1. PEDF stands for Pigment Epithelial Derived Factor which has been added into the introduction section as well as the full name for all the other biomarkers.
  2. We have now increased the size of the labels in Fig. 2.

Reviewer 2 Report

This is an interesting paper defining potential improvement to assessment of BCR using combined set of pre-operative biomarkers. The design of the study and statistical analysis are generally sound. However, one concern for this reviewer is the use of the combined cohort for the five-year assessment of BCR. The results from the combined cohort should be presented as a model for the future studies and not as final conclusion. One solution for the result presentation would be to state that insufficient number of subjects did not allow full analysis of biomarker performance at 5-year point, but the use the combined cohort allowed modeling of the performance showing an improvement. Prospective studies should be proposed for the biomarkers.

Author Response

Thank you for your suggestion. This was the reason why we included the data as supplementary and was not the main focus of the paper. We have made some changes to the Patients and Methods section to reflect your comment. We also updated the Discussion section proposing prospective studies for the biomarkers.

Reviewer 3 Report

The article titled, “Integrating Serum Biomarkers into Prediction Models for Bio-2 chemical Recurrence following Radical Prostatectomy” is well written and presented.

I agree with the authors views that more development is required before these findings could be adopted for practice.

There are a few typos, like on line 91- -80°C. The degree symbol should be superscript.

All the very best.

Author Response

We made this correction and undertaken further spell checking of the document.

Reviewer 4 Report

This manuscript that entitled “Integrating Serum Biomarkers into Prediction Models for Biochemical Recurrence following Radical Prostatectomy” aimed to identify serum biomarkers for prediction biochemical recurrence. There are some recommendations for authors.

Major concerns

All the results were merely based on clinical statistics, lacked experimental data. The authors picked up several serum proteins for biomarkers evaluation, but I did not observe any rationale for why these proteins should be investigated. In addition, CD14 was monocytes marker, the table 2 confused me by the concentration of CD14. Did CD14 secrete from cell membrane, and it might serve as a prediction factor of biochemical recurrence? This manuscript is simplistically to interpret purpose, results and discussion.

Author Response

  1. This is a retrospective study based on clinical samples collected from a treated cohort so as to build a model to predict patient outcome following patients’ follow up. The next study in the validation of this model would be a prospective randomised trial. We reemphasize this in the Discussion section. To investigate the clinical utility, we compared the experimental biomarker model with the current gold standard clinical model (NCCN).
  2. Thank you for this point, we have updated the introduction with a description of each biomarker selected giving insight into their role in cancer development.
  3. CD14 is released from the membrane and measured free in the serum.

Round 2

Reviewer 4 Report

The authors had provided rational explanations for all questions and the inadequacy of manuscript was improved. This manuscript seems to have reached the criterion for publication.